# Co-Expression Profile of TNF Membrane-Bound Receptors Type 1 and 2 in Rheumatoid Arthritis on Immunocompetent Cells Subsets

**DOI:** 10.3390/ijms21010288

**Published:** 2019-12-31

**Authors:** Alina Alshevskaya, Julia Lopatnikova, Julia Zhukova, Oksana Chumasova, Nadezhda Shkaruba, Aleksey Sizikov, Irina Evsegneeva, Victor Gladkikh, Aleksander Karaulov, Sergey V. Sennikov

**Affiliations:** 1Federal State Budgetary Scientific Institution Research Institute of Fundamental and Clinical Immunology (RIFCI), 630099 Novosibirsk, Russia; alkkina@yandex.ru (A.A.); lopatnikova_j_a@ngs.ru (J.L.); zhukova1982@rambler.ru (J.Z.); chumoks@mail.ru (O.C.); sen-nadezhda@yandex.ru (N.S.); depaidici@online.nsk.su (A.S.); 2Federal State Autonomous Educational Institution of Higher Education I.M. Sechenov First Moscow State Medical University of the Ministry of Health of the Russian Federation, 119435 Moscow, Russia; ivevsegneeva@yandex.ru (I.E.); drkaraulov@mail.ru (A.K.); 3Biostatistics and Clinical Trials Center, 630090 Novosibirsk, Russia; gladvs_ru@mail.ru; 4Novosibirsk State University, 630090 Novosibirsk, Russia

**Keywords:** TNF, cytokine receptors, co-expression, rheumatoid arthritis, disease activity

## Abstract

Introduction: Tumor necrosis factor (TNFα) is an important proinflammatory cytokine in rheumatoid arthritis (RA) immune processes. However, TNFα activity and functions may be regulated by soluble receptors, which act as decoys, and by number, density, and co-expression of its membrane-bound receptors type 1 and 2 (TNFR1 and TNFR2). The aim of this study was to reveal associations between TNFR1/2 co-expression profile parameters and RA disease activity indicators. Methods: PBMC were analyzed from 46 healthy donors and 64 patients with RA using flow cytometry. Patients were divided according to the disease activity score (DAS) 28 index into groups with high (*n* = 22, 34.4%), moderate (*n* = 30, 46.9%), and low (*n* = 12, 18.8%) disease activity. Co-expression of TNFR1 and TNFR2 was studied by evaluating the percentage of cells, with different receptors, and by counting the number of receptors of each type per cell, using QuantiBritePE beads. Associations between disease severity and activity indicators and parameters of TNFα receptor expression in subpopulations of immune cells were studied. Results: T cell subsets from RA patients were characterized by co-expression of TNFR1 and TNFR2, and were found to differ significantly compared with healthy donors. Memory cells both among T helper cells and cytotoxic T cells demonstrated the most significant differences in TNFR-expression profile. Multivariable logistic regression revealed model to identified RA patients from healthy individual based on the TNFR1/2 co-expression parameters. Conclusion: The profile of TNFR1\2 co-expression differs in RA comparing with health. Proportion of TNFR1+TNFR2- cells increased significantly among memory T helper cells and activated cytotoxic T cells, and decreased significantly among naïve cytotoxic T cells and T regulatory cells as compared with health. The parameters of TNFR1\2 co-expression in RA are associated with clinical and laboratory indicators of disease activity.

## 1. Introduction

The immunoregulatory cytokine tumor necrosis factor (TNF)-α plays a key role at all stages of the pathogenic process of rheumatoid arthritis (RA) [1,2]. TNFα functions only in the presence of a sufficient number of specific receptors on the cell surface. The interaction of cytokine with type 1 receptor ensures the development of inflammatory and cytotoxic reactions, and stimulates apoptosis. Signal transmission through the type 2 receptor usually increases the survival of cells and stimulates their proliferation, synthesis, and secretion of various mediators, including cytokines [3]. However, there are alternative possibilities for triggering signaling pathways through type 1 and type 2 receptors, which depend on the proportion of cells with receptors, expression density, co-expression on subpopulations, and the structure of the receptor. In particular, change in the ratio of different types of receptors can lead to a shift in the balance between pro-apoptotic and proliferative signaling pathways [4]. Changes in the expression of TNFα receptors during active inflammation may be triggered by the ongoing pathological processes. Moreover, previous studies demonstrate that a change in the expression level of type 1 and 2 receptors for TNFα in the main blood cell populations in immune-mediated diseases is associated with the activity indices of the pathological process [5,6]. For tuberculosis patients, increase of TNFR1 expression on immune cells as compared to healthy donors was found [5], and in atopic dermatitis TNFR1 and TNFR2 expression was found to increase in lymphocyte and monocyte populations as compared to health [6]. Thus, different immunopathogenetic processes leads to redistribution of TNFR1 and TNFR2 in immune cell subsets. However, to date, there are no data analyzing the co-expression of different types of TNFα receptors on individual subpopulations of cells actively involved in pathological processes (T regulatory cells, memory T cells, T helper cells).

We hypostasized that receptor co-expression may change significantly in pathology, influencing the potential of the immune system to respond to a particular type of trigger or to a therapy. The first step to test this hypothesis was to investigate the influence of disease activity and severity on receptor co-expression. The aim of the study was to analyze the expression of type 1 and 2 receptors for TNFα in individual subpopulations of peripheral blood cells in patients with RA compared with healthy donors. We also aimed to identify differences between TNFR1/2 co-expression on cell subpopulations actively involved in immunopathological processes and to reveal the trends in changing the parameter during disease flares.

## 2. Results

### 2.1. TNFR1/2 Co-Expression in RA Patients with Different Disease Activity

Preliminary studies of healthy donors revealed significant heterogeneity of expression and co-expression levels of type 1 and 2 receptors for TNFα on individual subpopulations of immune cells. Further to this, we studied the most significant subpopulations of lymphocytes actively involved in the pathogenic processes associated with RA, by analyzing co-expression of type 1 and 2 receptors for TNFα, and calculated the average number of receptors on the cell surface with each of the co-expression variants. The following populations were selected: total monocytes; total B lymphocytes; total T lymphocytes; subpopulations of cytotoxic T cells (CD8+), T helper cells (CD4+), activated CD8+ cells, activated CD4+ cells, subpopulations of memory T cells (CD45RO+), and naïve T cells (CD45RA+) among cytotoxic and T helper cells and T regulatory cells (Tregs, CD4+CD25 high CD127 low). 

We found that the expression of type 1 and 2 receptors for TNFα differ significantly on the surface of analyzed cell populations (monocytes, B cells, total T cells, and Tregs) between each receptor and compared with healthy donors (Figure 1).

The most pronounced differences in the co-expression profile of type 1 and 2 receptors for TNF were shown for B lymphocytes and cytotoxic T lymphocytes. There were significantly fewer B cells expressing only type 2 receptor compared with healthy donors. The proportion of cells with expression of the type 2 receptor varied from 9.4 to 23.8% in RA patients as compared to 50.8% in healthy donors. Conversely, cytotoxic T cells in patients with RA were characterized by an increase in the percentage of cells expressing the type 1 receptor with 13−23.6% of double positive cells (as compared with 7.2% in healthy donors) and 10−16.2% of cells with single expression of the type 1 receptor (as compared with 1.6% in healthy donors).

The differences between subpopulations of T helper cells (Figure 2) and cytotoxic T cells (Figure 3) were analyzed. The most pronounced redistribution of receptors was shown for subpopulations of T helper memory cells and cytotoxic T memory cells. More than 90% of healthy donor memory T helper cells and memory cytotoxic T cells expressed type 2 receptors and less than 2% of the cells were negative for TNF receptors. Concurrently, from 12 to 54.2% of RA patient’ cells (depending on disease activity) were negative for both type 1 and 2 receptors for TNFα. Type 1 receptor expression was predominantly expressed by healthy donor cytotoxic naïve T cells, while the type 2 receptor predominated on cytotoxic T cells from patients with RA.

The obtained data demonstrated that distribution of type 1 and type 2 receptors for TNFα differs significantly between immune cell populations and changed during pathology. The highest percentage of double-negative cells in healthy donors was naive T helper cells and the highest percentage of double-positive cells was activated cytotoxic lymphocytes. Furthermore, the largest percentage of cells expressing the type 1 receptor alone was cytotoxic memory T lymphocytes, while the largest percentage of cells expressing the type 2 receptor alone was naive cytotoxic T lymphocytes and memory T helper cells. In RA patients, there was a tendency for an increase in the percentage of double-negative cells among most of populations, which may be due to the active process of shedding of the membrane-bound receptors. Additionally, there was a redistribution of receptors in the subpopulations. However, the largest percentage of cells expressing only type 1 receptor was cytotoxic memory T lymphocytes, similar to healthy donors.

The mean number of receptors per cell were calculated for each subset (Figure 4 and Table 1). The most pronounced differences in quantitative expression of the receptors were found for memory cells both among T helper cells and cytotoxic T cells: the number of type 2 receptors on these subpopulations in patients with RA exceeded that in healthy patients 5.1-fold and 4.7-fold, respectively.

### 2.2. Correlations between Co-Expression Parameters and Disease Severity Indicators

A total of seven indicators of receptor expression were identified that differ significantly in RA patients with high disease activity compared with healthy donors but do not differ between healthy donors and patients with low disease activity: four indicators based on the number of receptors on cells (number of type 2 receptors on double-positive TNFR1+TNFR2+ cells among B cells and activated cytotoxic lymphocytes (CD8+CD25+), number of type 2 receptors among TNFR1−TNFR2+ B lymphocytes, number of type 1 receptors among TNFR1+TNFR2− monocytes) and three indicators based on the percentage of cells expressing certain combinations of receptors (percentage of TNFR1+TNFR2− cells among the total pool of T lymphocytes and percentage of cells expressing at least one of the two types of receptors for TNFα among naive T helper cells and memory T helper cells). These indicators may be promising for the establishment of threshold model to identify patients who have the potential to for their disease to flare up.

A correlation analysis of the relationship between disease severity and activity indicators (DAS-28 index, disease duration, radiological stage, activity stage, rheumatoid factor (RF) levels, anti-CCP and CRP levels, the presence of systemic manifestations and erosive arthritis) and parameters of TNFα receptor expression was performed in the subpopulations of immune cells (Table 2). Roentgenology stage and presence of systemic features were not found to correlate with any TNFR1 or TNFR2 expression parameter. 

### 2.3. Regression Analysis for Disease Activity Indicators

As several co-expression indicators were found to be associated with disease severity and activity, univariable and multivariable regression analysis were performed to find possible predictors for RA (through all RA patients and healthy individuals), for high disease activity (through all RA patients and healthy individuals), and for low disease activity (through RA patients) (Table 3). *R*^2^ were 0.68 for all RA patients, 0.32 for high disease activity, and 0.3 for low disease activity. Thus, parameters of TNFR1/2 co-expression may be used as predictors to identified RA patients from healthy individuals.

## 3. Discussion

We investigated the expression of TNFα receptors on immune cells in RA patients, and found that the distribution pattern of TNFα receptors on immune cells differs between patients with RA and healthy donors. Specifically, the total proportion of cells expressing at least one of the receptors decreased significantly on B cells and memory T helper cells, and increased significantly on naive cytotoxic T cells, naive T helper cells, monocytes, and T regulatory cells compared with healthy donors. This indicator reflects the overall sensitivity of the cell subpopulation to the action of cytokine, and, accordingly, the ability of the cells to react to a change in TNF concentration.

One of the most interesting features revealed by this study was the change in the distribution pattern among patients with different RA activity levels. Although the indicators and the pattern of receptor distribution in RA patients significantly differed from healthy, in patients with low disease activity there was no tendency for the indicators to approach the level of healthy donors. This feature was revealed for the first time. We believe that this indicates that with a decrease in the activity of RA, processes of stabilization of the activity of the immune system occur; however, the new conditionally stable system differs greatly from that of healthy donors. Further studies of the state of patients in the dynamics of the disease are needed to confirm this hypothesis.

The functional activity of T regulatory cells and their participation in inflammatory processes are compromised in RA [7]. A number of authors associate these changes to the action of TNF, which results in inhibition of the suppressor function of T regulatory cells [8]; this process is associated with the expression of the TNF type 2 receptor [9]. We observed a redistribution of these receptors in RA, namely, a tendency in increasing the proportion of TNFR1−TNFR2+ cells with a significant decrease in the proportion of double-positive cells. At the same time, the average number of type 2 receptors on the cell surface was almost 2 times higher in patients with RA than in healthy donors. The role of type 2 receptors in providing suppressor functions of immune cells has been previously shown. We suggest that the increase in expression of type 2 receptors detected on T regulatory cells is associated with a change in their suppressor activity in RA.

Reduced expression of type 2 receptors on B cells and cytotoxic T cells in patients with RA may also be associated with changes in their suppressor activity [10]. Meanwhile, the redistribution of receptors in these populations may occur in different ways. In the case of B cells, this reduction is manifested by a decrease in the proportion of TNFR2+ cells and a decrease in the number of type 2 receptors on the cells. The proportion of receptor-positive cytotoxic T cells decreased, while the number of receptors on them did not change or even increased slightly. These changes in B cells led to their decreased sensitivity to TNF and the effects mediated by type 2 receptors, while redistribution of receptors in cytotoxic T cells and an increase in their number may represent a compensatory mechanism.

We have demonstrated that the proportion of TNFR2+ cells (and, in general, cells with receptors for TNF) among memory T helper cells is sharply reduced in patients with RA compared with healthy donors. At the same time, these cells are characterized by the highest quantitative expression of type 2 receptors among all the studied subpopulations. Such a sharp increase in the density of receptor expression on individual cells may be due to a change of their functional ability.

An interesting variant of receptor redistribution was demonstrated for naive cytotoxic T cells compared with the healthy donors: the proportion of cells with type 1 receptors was dramatically increased, while the percentage of cells expressing type 2 receptors was sharply reduced. Since type 1 and type 2 receptors implement different functions, these changes may indicate a change in the nature of the response of these cells to cytokines [11,12], and as a result, a change in their functional activity in RA. Evidence suggests that triggering a signal through type 1 and type 2 receptors simultaneously has significant therapeutic potential. In a model of autoimmune encephalitis [13], the different roles of type 1 and type 2 receptors for TNF have been demonstrated. For type 2 receptors, their ability to induce pronounced inflammation was shown, with minimal signs of demyelination; and on the contrary, the type 1 receptors, although they did not contribute to inflammation, actively participated in the destruction of the nervous system’s autoimmune processes. This example shows that the redistribution of receptors on a subpopulation of cells favoring the expression of TNFR2 may be a defensive reaction or a manifestation of the aggressive course of the auto-inflammatory process in RA; however, this requires further investigation.

We studied the associations between disease severity and activity indicators (DAS-28 index, disease duration, radiological stage, activity stage, RF levels, anti-CCP and CRP levels, the presence of systemic manifestations and erosive arthritis) and parameters of TNFα receptor expression in immune cells. A series of correlated relationships were identified that led to study of predicative models using parameters of TNFR1/2 co-expression. Model aimed to define RA patients with high or low disease activity showed low predicative potential. However, model, identifying RA patients through hole cohort of individual showed 68% *R*^2^. Indicators from this model may have diagnostic value to identify RA patients.

Although our study demonstrates for the first time the relationship between TNFR1/2 co-expression parameters and the clinical characteristics of patients with rheumatoid arthritis, it has limitations. Firstly, this is a small sample of subgroups of patients with different disease activity, which did not allow revealing statistically significant differences in a number of trends. Secondly, it is the sex-age heterogeneity of patients and healthy donors which could affect the characteristics of the distribution of receptors in subpopulations.

Our study raised a number of questions for further in-depth study of the relationship between co-expression and clinical parameters. Firstly, the effect is of considerable interest that the indicators of patients with low disease activity in most cases are not intermediate between high RA activity and healthy donors, but on the contrary are very different. This testifies to the fact that, even with effective therapy, the receptor system forms a certain balance that differs from healthy donors’ one. Therefore, for an effective clinical response, it is not always necessary to strive to bring indicators with the norms of conditionally healthy donors. Secondly, the established associations between clinical indicators and expression indices make it possible to suggest the presence of feedback. Namely, the use of targeted modulation of the receptor level to change the level of inflammatory processes in animal models of the disease, followed by implementation in clinical practice is of interest for further studies.

## 4. Materials and Methods

### 4.1. Samples

Mononuclear cells (MNC) were isolated from the peripheral blood of healthy donors and patients with RA. Health group included donors from Novosibirsk Blood Station who provided their written informed consent to participate the study. RA group included undergoing treatment at the Rheumatology Department of the Clinic of Immunopathology of the Scientific Research Institute of Fundamental and Clinical Immunology (RIFCI). Diagnosis of RA was estimated according to ACR/EULAR (2010) Classification Criteria for RA [14]. The study was approved by the local ethics committee of RIFCI (protocol no. 24, dated 8 September 2016). The study included 46 healthy donors aged 18–77 years (median (interquartile range; IQR), 36.5 (30:54) years) including 16 (34.8%) males, 30 (65.2%) females and 64 patients with RA aged 22–83 years (median (IQR), 55 (45:65) years), among which 54 (85.4%) were women. The demographic and clinical characteristics of the patients and healthy donors are presented in Table 4. Our group differed significantly by both age (*p* = 0.005) and gender (*p* = 0.094); however, these indicators were not found to be associated with any studied parameters of receptor expression. RA patients received main therapy as followed: rituximab (3 (4.7%)), rituximab+sulfasalazine (2 (3.1%)), rituximab+leflunomide (9 (14.1%)), rituximab+methotrexate (14 (21.9%)), methotrexate (10 (15.6%)), methotrexate+oral glucocorticoids (10 (15.6%)), leflunomide+methotraxate (1 (1.6%)), oral glucocorticoids (6 (9.4%)), tocilizumab (4 (6.3%)), abatacept+methotraxate (1 (1.6%)), and leflunomide (4 (6.3%)). No associations between therapy and studied expression parameters were found. 

Venous blood (6 mL) was collected on an empty stomach from the ulnar vein under sterile conditions into vacuum tubes with K3-EDTA anticoagulant (ethylenediaminetetraacetic acid tripotassium salt, Vacuette K3-EDTA, Greiner Bio-One GmbH, Kremsmünster, Austria).

Sample preparation was performed using BD FACS Lysing Solution buffer (cat. no. 349202; BD, San Jose, CA, USA) according to the manufacturer’s instructions.

### 4.2. Flow Cytometry

Evaluation of phenotypic characteristics was performed by flow cytometry (FACSVerse cytometer (BD, USA)) using monoclonal antibodies: anti-human CD3 APC/Cy7, anti-human CD19 PE/Cy7, anti-human CD8 APC/Cy7, anti-human CD14 PerCP, anti-human CD25 FITC, anti-human CD127 (IL-7Rα) APC/Cy7, anti-human CD45RO FITC, anti-human CD45RA Pacific Blue, anti-human CD45 PerCP, anti-human CD4 Pe/Cy7, anti-human TNFRI-PE, anti-human TNFRII-PE, anti-human TNFRI-APC, and anti-human TNFRII-APC (R&D Systems, Minneapolis, MN, USA). Data processing and calculation of fluorescence intensity values were performed using FACSDiva 7 software (BD, USA).

To obtain the calibration curve and convert the fluorescence intensity values into absolute number of receptors for cells expressing the corresponding marker, BD QuantiBRITE PE kit (BD Biosciences, San Jose, CA, USA) containing four fractions of lyophilized beads, each carrying a different level of phycoerythrin (PE), was used. According to the manufacturer’s instructions, a log–log graph of the number of PE molecules versus fluorescence intensity was plotted based on the results of bead analysis, and linear relationship was identified using the trend line. The obtained relationship was used to determine a formula for converting the fluorescence intensity values for the PE channel into the number of PE molecules for each of the studied subpopulations, and the average values of the number of receptors per cell were calculated.

For the simultaneous determination of the number of TNFα receptors types 1 and 2 on cells in all four fractions of each cell line (TNFR1+TNFR2−, TNFR1+TNFR2+, TNFR1−TNFR2+, TNFR1−TNFR2−), double labeling of paired samples was carried out. Each sample with a certain dose of rhTNF (or control sample without TNF) was divided into two tubes stained with TNFR1-PE and TNFR2-APC or with TNFR2-PE and TNFR1-APC.

After cytometry analysis, the number of type 1 receptors (for TNFR1+TNFR2− and TNFR1+TNFR2+ fractions) was calculated in tubes with TNFR1-PE and TNFR2-APC. The number of type 2 receptors (for TNFR1−TNFR2+ and TNFR1+TNFR2+ fractions) was calculated in tubes with TNFR2-PE and TNFR1-APC. The percentage of each fraction was determined as the mean between the two samples.

Detailed gating strategy and examples of co-expression analysis were provided in our previous publication describing this analysis in healthy individuals [15].

### 4.3. Statistics

Statistical data processing was performed using STATISTICA 7.0 software (StatSoft, Tulsa, OK, USA). The data are presented as median and IQR. Independent samples were compared by determining the statistical significance of differences using the non-parametric Kruskal–Wallis test with multiple comparison of medians (comparison of identical indicators for different subpopulations and identification of differences between the studied subgroups). Correlations between the studied parameters were determined using Pearson correlation coefficient (at *p* < 0.05). Univariable and multivariable logistic regression were performed. Multifactor models were obtained by reducing insignificant indicators to achieve the best model quality indicators (lowest AIC and highest *R*^2^). The statistical significance of changes, in parameters in dependent samples in patients after correction of background therapy, was established using Wilcoxon test (differences were considered significant at *p* < 0.05).

## 5. Conclusions

In the present study, we showed that a redistribution of type 1 and type 2 receptors for TNFα takes place in RA immune cells compared with healthy donors. These changes are associated with clinical and laboratory indicators of disease activity.

## Figures and Tables

**Figure 1 ijms-21-00288-f001:**
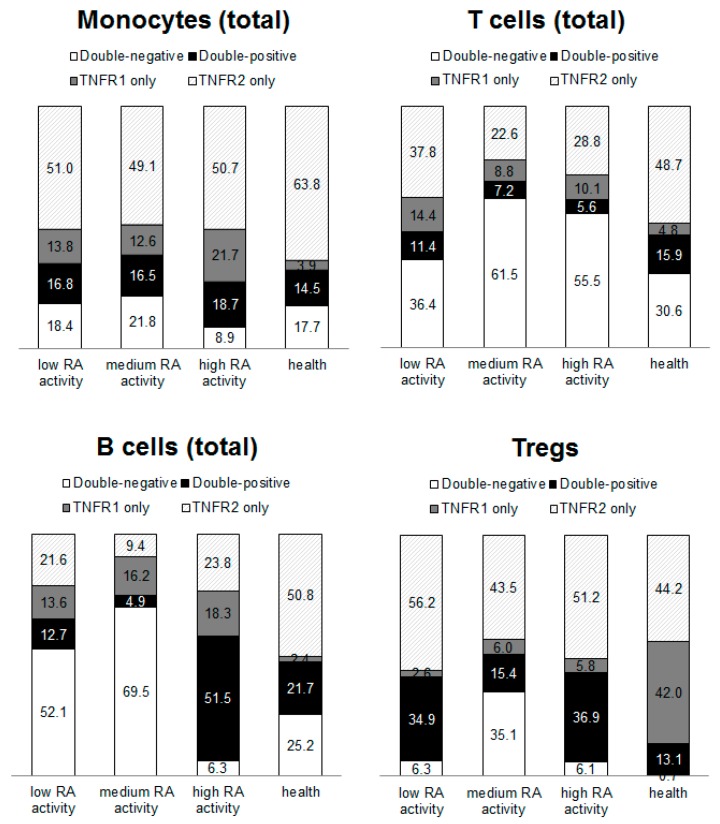
Co-expression profiles for TNF receptors in main subpopulations of MNC PC of healthy donors and patients with RA.

**Figure 2 ijms-21-00288-f002:**
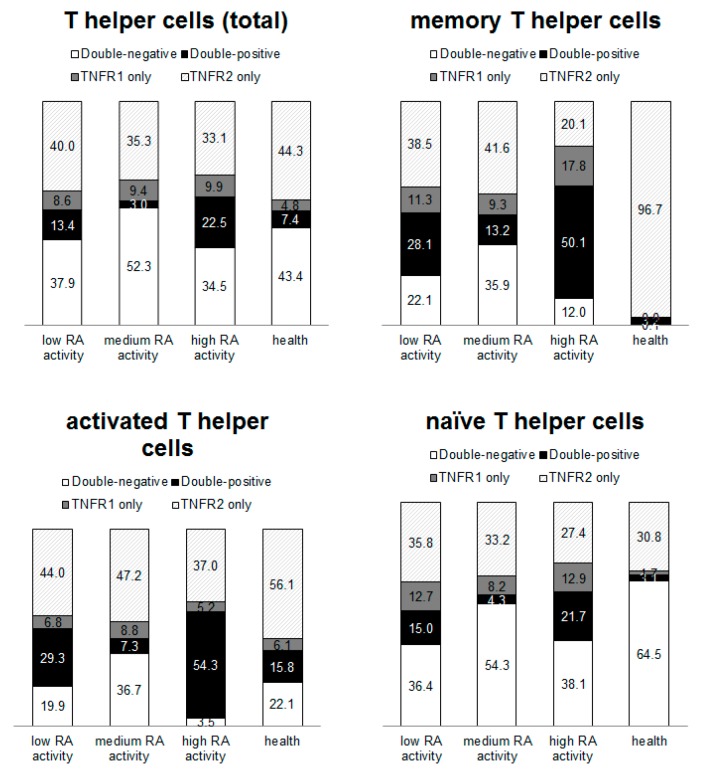
Subpopulations of T helper cells. Proportion of cells expressing type 1 and 2 receptors for TNFα among T helper cells. Data have been normalized to the total median value.

**Figure 3 ijms-21-00288-f003:**
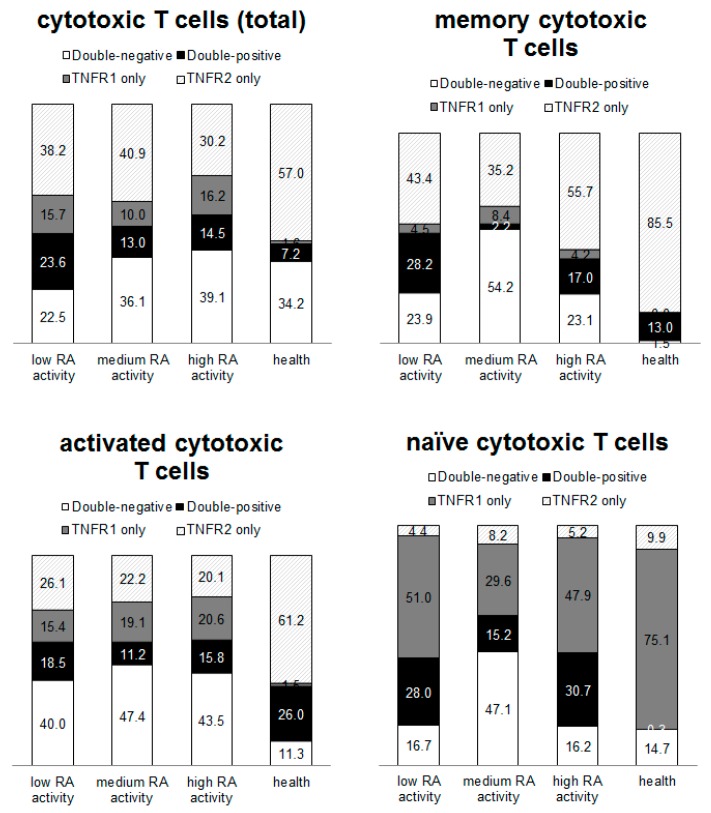
Subpopulations of cytotoxic T cells. Proportion of cells expressing type 1 and 2 receptors for TNFα among cytotoxic T cells. Data have been normalized to the total median value.

**Figure 4 ijms-21-00288-f004:**
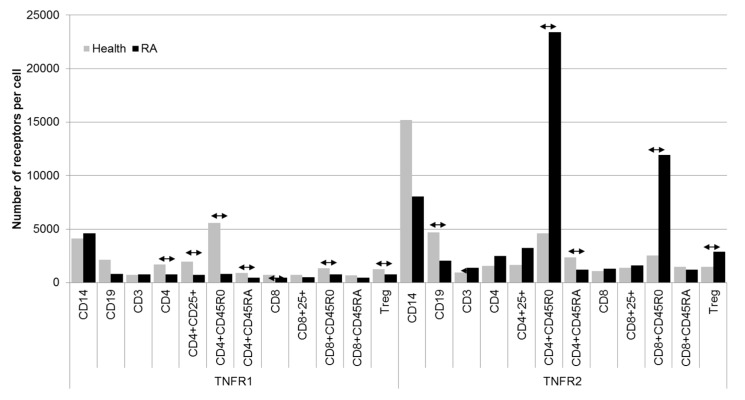
Number of type 1 and 2 receptors for TNFα on the surfaces of immune cells. Data are presented as median. Arrows indicate statistically significant differences (*p* < 0.05).

**Table 1 ijms-21-00288-t001:** Number of type 1 and 2 receptors for TNFα on the surfaces of immune cells. Data are presented as median and interquartile range.

TNFR1/2	Subset	Health	RA	*p* Value
TNFR1	CD14+	4107 (2713–12,435)	4604 (1853−10,353)	0.640
	CD19+	2147 (1510–6077)	820 (577−2218)	0.757
	CD3+	743 (190–2036)	754 (481−1844)	0.120
	CD4+	1685 (673−5286)	784 (437−1074)	<0.001
	CD4+CD25+	1965 (973−14,891)	730 (500−970)	0.000
	CD4+CD45R0+	5575 (1893−12,654)	818 (530−1142)	<0.001
	CD4+CD45RA+	918 (276−2269)	469 (376−674)	0.000
	CD8+25+	729 (127−1885)	450 (366−525)	0.034
	CD8+CD2525+	703 (164−1658)	523 (415−619)	0.203
	CD8+CD45R0+	1336 (292−3363)	787 (565−1008)	0.047
	CD8+CD45RA+	667 (175−1489)	443 (404−681)	0.178
	Treg	1253 (569−6191)	764 (592−1560)	0.020
TNFR2	CD14+	15181 (12,588−33,641)	8029 (4809−21,546)	0.785
	CD19+	4704 (3155−12,446)	2056 (1264−3022)	0.033
	CD3+	935 (120−2269)	1404 (1075−2405)	<0.001
	CD4+	1570 (463−3504)	2477 (1434−4620)	0.475
	CD4+CD25+	1657 (465−3750)	3238 (1586−6582)	0.327
	CD4+CD45R0+	4601 (2547.5−12,913.5)	23392 (3536−56,201)	0.048
	CD4+CD45RA+	2372.5 (979−6276.5)	1209 (970−1981)	0.009
	CD8+25+	1092 (322−2528)	1302 (833−2781)	0.252
	CD8+CD2525+	1370 (388−4578)	1628 (922−3060)	0.728
	CD8+CD45R0+	2543.5 (1146.5−7057)	11,915 (2476−27,212)	0.009
	CD8+CD45RA+	1496.5 (336.5−3657)	1220 (857−2415)	0.743
	Treg	1460 (201−5029)	2885 (1790−6996)	0.237

**Table 2 ijms-21-00288-t002:** Correlation analysis (*p* < 0.05).

Cell Subset	DAS-28	RA Duration	RF Level	ACCP	CRP Level	Erosive Arthritis	Active Stage
CD14	% T1 (r = −0.72) % T2 (r = 0.82) N T2++ (r = 0.71) N T2+ (r = 0.71)	-	% T2 (r = 0.78)	-	%-- (r = 0.97) % T1 (r = 0.79)	-	% T2 (r = 0.78)
CD19	-	% T1 (r = 0.77)	-	-	N T2++ (r = 0.96) N T2+ (r = 0.95) N T2 full (r = 0.95)	-	-
CD3	-	% T1 (r = 0.71)	% T2 (r = 0.70)	-	N T2+ (r = 0.77) N T2 full (r = 0.77)	-	-
Treg	-	% ++ (r = 0.82) N T1 + (r = 0.87) N T2++ (r = 0.89) N T2full(r = 0.94) N T2+ (r = 0.93)	N T1++ (r = 0.78)	-	-	-	-
CD4	-	% ++ (r = 0.75)	-	-	-	-	-
CD8	-	% ++ (r = 0.79) % T1 (r = 0.81) % T2 (r = −0.73)	-	-	-	-	-
CD4 activated	-	% ++ (r = 0.85)	-	% T1 (r = −0.72)	N T2 ++ (r = 0.72)	-	-
CD4 memory	-	-	% ++ (r = 0.73) NT2++(r = 0.74) N T2full r = 0.79) N T2 + (r = 0.80)	% T1 (r = −0.96)	-	N T1 ++ (r = −0.73) N T1 full (r = −0.73)	-
CD4 naïve	-	-	N T2full (r = 0.75) N T2+ (r = 0.73)	-	-	% T1 (r = −0.86)	-
CD8 memory	N T2full (r = 0.75) N T2 + (r = 0.76)	% ++ (r = 0.90) % T2= (r =−0.074) N T1 ++(r = 0.77) N T1full(r = 0.77)	N T2++(r = 0.78) N T2full(r = 0.90) N T2 +(r = 0.98)	% T1+ (r = −0.89)	-	-	N T2 + (r = 0.70)
CD8 naïve	-	% ++ (r = 0.83) N T ++ (r = 0.77) N T1full(r = 0.78)	-	-	-	-	-

%--: percentage of double-negative cells. %++: percentage of double-positive cells. %T1: percentage of TNFR1+TNFR2- cells. %T1: percentage of TNFR1-TNFR2+ cells. N T1 (T2) ++: number of TNFR1 (TNFR2) on double-positive cells. N T1+: number of TNFR1 on TNFR1+TNFR2- cells. N T2+: number of TNFR2 on TNFR1-TNFR2+ cells. N T1 (T2) full: total number of TNFR1 (TNFR2) in subset.

**Table 3 ijms-21-00288-t003:** Multivariable logistic regression analysis.

Indicator		OR (2.5–97.5% CI)	*p*-Value	*R* ^2^
High disease activity RA patients	Number of TNFR2 on TNFR2+ cells from CD8CD45R0 cells	1.002 (1.001–1.005)	0.016	0.32
	Percentage of TNFR1+TNFR2− cells from CD8 cells	1.41 (1.08–1.96)	0.021	
	Percentage of TNFR1+TNFR2− cells from activated CD8 cells	0.75 (0.54–0.98)	0.05	
RA (vs. health)	Percentage of TNFR1+TNFR2− cells from CD4CD45RA cells	1.326 (1.111–1.695)	0.008	0.68
	Number of TNFR1 on TNFR1+ cells from CD4CD45R0 cells	0.998 (0.995–1)	0.005	
	Number of TNFR2 on TNFR2+ cells from CD3 cells	1.003 (1.001–1.008)	0.008	
Low disease activity (vs. medium and high)	Percentage of double-positive cells from Tregs	1.039 (1–1.08)	0.044	0.3
	Percentage of double-negative cells from CD8 cells	1.059 (1–1.13)	0.045	
	Percentage of TNFR1-TNFR2+ cells from CD8CD45RA cells	1.067 (1.01–1.15)	0.041	

**Table 4 ijms-21-00288-t004:** Baseline characteristics of the studied groups.

Indicator		Healthy Donors (*n* = 43)	RA Patients (*n* = 64)
Gender	Females, *n* (%)	30 (65.2%)	54 (84.4%)
Age	Median (IQR)	36.5 (30: 54)	55 (45: 65)
Disease duration	Median (IQR)	-	8.5 (5: 17)
Erosive arthritis	*n* (%)	-	52 (81.3%)
Systemic manifestations of arthritis	*n* (%)	-	32 (50%)
Radiological stage	I, *n* (%)	-	1 (1.6%)
	II, *n* (%)		25 (39.1%)
	III, *n* (%)		30 (46.9%)
	IV, *n* (%)		8 (12.5%)
Degree of activity	0, *n* (%)	-	7 (10.9%)
	1, *n* (%)		4 (6.3%)
	2, *n* (%)		32 (50%)
	3, *n* (%)		21 (32.8%)
	4, *n* (%)		0 (0%)
DAS-28 disease activity	low (<3.2), *n* (%)	-	12 (18.8%)
	medium (3.2−5.1), *n* (%)		28 (43.8%)
	high (>5.1), *n* (%)		24 (37.5%)

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
