# Peer review of "Co-Expression Profile of TNF Membrane-Bound Receptors Type 1 and 2 in Rheumatoid Arthritis on Immunocompetent Cells Subsets"

_ijms, 2019, doi:10.3390/ijms21010288_

Round 1
Review 1 Report
Alina et al. describe the expression profile of TNFα receptors, TNFR1 and TNFR2, on immune cells, including their individual subpopulations, in patients with rheumatoid arthritis (RA) compared with healthy donors, and show the changes in the distribution pattern of TNFα receptors (TNFR1/2) in the patients with different disease activity levels. Although these contain some interesting findings, I think there are many ambiguous and difficult parts to understand throughout the manuscript.
Comments and concerns:
How did the authors diagnose the patients with RA (What classification criteria did the authors use to diagnose the patients)? (Page 5, Line 3-5) The sentence, “the proportion of cells with expression of the type 2 receptor was 2 times less (27.6% versus 68.7%)” is unclear. Does the “cells” mean B cells(total)? If so, which parts of figure 1 (B cells) are 6%and 68.7%? (Page 5, Line 5-8: Conversely, cytotoxic T cells in patients with RA were …) Are these descriptions of “cytotoxic T cells” shown somewhere in figure 3? (Page 5, Line 15-17) Here, the authors describe the results for all patients with RA. Therefore, it would be helpful to understand if the authors show the results of the whole patients with RA, combining all patients with low to high disease activity into one graph (100% stacked bar), in the figures (Figs.1~3). (Fig 2&3) I think that it would be better to show the proportion of memory T helper cells, naïve T helper cells, and activated T helper cells in the total T helper cell population (similarly for cytotoxic T cells in Figure 3). (Fig 2&3) Although the patients with low and high disease activity tend to have similar patterns of TNFR expression on T helper cells (as well as on memory and naïve cytotoxic T cells), the patients with medium disease activity have different patterns of TNFR expression. How do the authors interpret these results? Have the authors examined changes in TNFR1/2 expression patterns on each subpopulation of immune cells before and after treatment? Lastly, how do the authors think the results of this study can be used in clinical practice?
Minor point:
typo (Page 9, Line 13: 216) healthy individuales→individuals
Author Response
How did the authors diagnose the patients with RA (What classification criteria did the authors use to diagnose the patients)? (Page 5, Line 3-5)
Answer: Information and citation was added, highlighted by yellow.
The sentence, “the proportion of cells with expression of the type 2 receptor was 2 times less (27.6% versus 68.7%)” is unclear. Does the “cells” mean B cells(total)? If so, which parts of figure 1 (B cells) are 6%and 68.7%? (Page 5, Line 5-8: Conversely, cytotoxic T cells in patients with RA were …) Are these descriptions of “cytotoxic T cells” shown somewhere in figure 3?
Answer: Full fragment was rewritten for better understanding, highlighted by yellow.
(Page 5, Line 15-17) Here, the authors describe the results for all patients with RA. Therefore, it would be helpful to understand if the authors show the results of the whole patients with RA, combining all patients with low to high disease activity into one graph (100% stacked bar), in the figures (Figs.1~3).
Answer: Indeed, in the original version of the article, we decided to describe the data on the RA patient population as a whole. However, after a thorough study and identification of differences between groups of patients with different RA activity, we now consider it incorrect to present the combined data. We rewrote fragments of the text that described the entire cohort of RA patients (highlighted by yellow)
(Fig 2&3) I think that it would be better to show the proportion of memory T helper cells, naïve T helper cells, and activated T helper cells in the total T helper cell population (similarly for cytotoxic T cells in Figure 3).
Answer: The determination of the percentage of sub-populations of full T cell population was not included in the study, since in preliminary experiments no clear associations were established between them and the level of co-expression of receptors.
(Fig 2&3) Although the patients with low and high disease activity tend to have similar patterns of TNFR expression on T helper cells (as well as on memory and naïve cytotoxic T cells), the patients with medium disease activity have different patterns of TNFR expression. How do the authors interpret these results?
Answer: Indeed, this fact is of significant interest. For most indicators of co-expression, we were not able to establish clear associations with an increase in disease activity, and patients with medium disease activity often have different indicators. We attribute this to the fact that this subgroup of patients differed in a number of parameters from high and low activity. In particular, with comparable age of patients in all three subgroups, among patients with moderate activity there was the largest proportion of patients with arthritis lasting no more than 5 years and a slightly larger proportion of patients receiving rituximab. However, no significant associations with the duration of the disease or therapy was established.
Have the authors examined changes in TNFR1/2 expression patterns on each subpopulation of immune cells before and after treatment?
Answer: Blood sampling in patients was limited to 7-12 days of hospital stay of patients. Unfortunately, in most patients during this time it was not possible to fix a significant change in the state and index of activity of the disease, therefore, repeated blood sampling was performed only in selected cases, which did not allow for a full-fledged statistical analysis.
Lastly, how do the authors think the results of this study can be used in clinical practice?
Answer: A section on further application of the results was added at the end of the discussion, highlighted by yellow.
Minor point: typo (Page 9, Line 13: 216) healthy individuales→individuals
Answer: Error was corrected, highlighted by yellow.
Answer: Section was added at the end of the discussion, highlighted by blue.
Reviewer 2 Report
The focus of this study was looking at TNFR1 and 2 expression in human immune cell populations in RA and control subjects and secondarily determine associations with disease severity and/or activity. This is an interesting outcome measure because if feasible, it would perhaps allow for an objective way of determining if a pt has mild, moderate or severe disease. While the 2nd aim was not met, it was able to identify RA subjects from healthy controls. However, are these expression levels related only to RA subjects or other subjects with autoimmune disease (ie lupus subjects or IBD subjects (where TNF plays a primary role)). Of note, there were differences in sex and age between RA and HD and although the authors do not believe this would affect TNFR expression, there are studies suggesting that TNF expression is mediated by estrogen levels so this should be discussed in the conclusions as a possible limitation of the study and in future studies, age/sex linked matches should be attempted.
Author Response
The focus of this study was looking at TNFR1 and 2 expression in human immune cell populations in RA and control subjects and secondarily determine associations with disease severity and/or activity. This is an interesting outcome measure because if feasible, it would perhaps allow for an objective way of determining if a pt has mild, moderate or severe disease. While the 2nd aim was not met, it was able to identify RA subjects from healthy controls. However, are these expression levels related only to RA subjects or other subjects with autoimmune disease (ie lupus subjects or IBD subjects (where TNF plays a primary role)).
Answer: Further studies in a large cohort of patients with various immuno-mediated diseases are needed to answer this question. The main aim of the current study, indeed, was to make sure that differences exist and are associated with clinical parameters.
Of note, there were differences in sex and age between RA and HD and although the authors do not believe this would affect TNFR expression, there are studies suggesting that TNF expression is mediated by estrogen levels so this should be discussed in the conclusions as a possible limitation of the study and in future studies, age/sex linked matches should be attempted.
Round 2
Reviewer 1 Report
Comments:
In “Abstract”, the authors state that "The profile of TNFR1/2 co-expression is altered during the course of RA”. Where does this result appear in the manuscript? In “Abstract”, I understand that there were differences in TNFR-expression profiles between healthy donors and RA patients, such as in memory cells, but I think it would be better to describe specifically what the differences were. (Legend for Figures 2 and 3) The authors state that "Proportion of cells expressing type 1 and 2 receptors for TNFα among T helper cells (or cytotoxic T cells) compared with total T cells" What does this "total T cells" mean (Do figure 2 and 3 have the same or different meanings)?
Author Response
In “Abstract”, the authors state that "The profile of TNFR1/2 co-expression is altered during the course of RA”. Where does this result appear in the manuscript? In “Abstract”, I understand that there were differences in TNFR-expression profiles between healthy donors and RA patients, such as in memory cells, but I think it would be better to describe specifically what the differences were. (Legend for Figures 2 and 3)
Answer: abstract section was modified, highlighted in yellow.
The authors state that "Proportion of cells expressing type 1 and 2 receptors for TNFα among T helper cells (or cytotoxic T cells) compared with total T cells" What does this "total T cells" mean (Do figure 2 and 3 have the same or different meanings)?
Answer: Figures legend were corrected.
Reviewer 2 Report
In the part of abstract, some abbreviation should show full name. In the part of intruction, the importance and different expression of TNFR type 1 or 2 should be disccused from previous studies. Different and numerous expresions of TNFR type 1 or 2 were reported in this stuy, and an adequate sumarized talbe may be made to increase the reading degree for readers. How about the different immunotherapy in these RA patients ? biologic therapy or DMARDs therapy may change the expression of TNFR in different cells In the part of conculsion, adequate statements about this study should be given for readers
Author Response
In the part of abstract, some abbreviation should show full name.
Answer: abstract section was modified, highlighted in yellow.
In the part of intruction, the importance and different expression of TNFR type 1 or 2 should be disccused from previous studies.
Answer: information was added.
Different and numerous expresions of TNFR type 1 or 2 were reported in this stuy, and an adequate sumarized talbe may be made to increase the reading degree for readers.
Answer: table was added.
How about the different immunotherapy in these RA patients ? biologic therapy or DMARDs therapy may change the expression of TNFR in different cells In the part of conculsion, adequate statements about this study should be given for readers
Answer: information about therapy were added into Methods section. No associations between therapy and studied expression parameters were found.